# Phenotypical Screening of an MMV Open Box Library and Identification of Compounds with Antiviral Activity against St. Louis Encephalitis Virus

**DOI:** 10.3390/v15122416

**Published:** 2023-12-13

**Authors:** Giuliana Eboli Sotorilli, Humberto Doriguetto Gravina, Ana Carolina de Carvalho, Jacqueline Farinha Shimizu, Marina Alves Fontoura, Talita Diniz Melo-Hanchuk, Artur Torres Cordeiro, Rafael Elias Marques

**Affiliations:** 1Brazilian Biosciences National Laboratory (LNBio), Brazilian Center for Research in Energy and Materials (CNPEM), Campinas 13083-100, Brazil; gsotorilli@gmail.com (G.E.S.); gravinahd@gmail.com (H.D.G.); ana.carvalho@lnbio.cnpem.br (A.C.d.C.); jacqueline.shimizu@lnbio.cnpem.br (J.F.S.); marina.fontoura@lnbio.cnpem.br (M.A.F.); talita.hanchuk@lnbio.cnpem.br (T.D.M.-H.); artur.cordeiro@lnbio.cnpem.br (A.T.C.); 2Department of Genetics, Microbiology and Immunology, Institute of Biology, State University of Campinas (UNICAMP), Campinas 13083-100, Brazil; 3Department of Cellular and Structural Biology, Institute of Biology, State University of Campinas (UNICAMP), Campinas 13083-100, Brazil

**Keywords:** St. Louis encephalitis virus, cell culture-based screening, drug repurposing, Flavivirus, cationic amphiphilic compounds (CAD)

## Abstract

St. Louis encephalitis virus (SLEV) is a neglected mosquito-borne Flavivirus that may cause severe neurological disease in humans and other animals. There are no specific treatments against SLEV infection or disease approved for human use, and drug repurposing may represent an opportunity to accelerate the development of treatments against SLEV. Here we present a scalable, medium-throughput phenotypic cell culture-based screening assay on Vero CCL81 cells to identify bioactive compounds that could be repurposed against SLEV infection. We screened eighty compounds from the Medicines for Malaria Venture (MMV) COVID Box library to identify nine (11%) compounds that protected cell cultures from SLEV-induced cytopathic effects, with low- to mid-micromolar potencies. We validated six hit compounds using viral plaque-forming assays to find that the compounds ABT-239, Amiodarone, Fluphenazine, Posaconazole, Triparanol, and Vidofludimus presented varied levels of antiviral activity and selectivity depending on the mammalian cell type used for testing. Importantly, we identified and validated the antiviral activity of the anti-flavivirus nucleoside analog 7DMA against SLEV. Triparanol and Fluphenazine reduced infectious viral loads in both Vero CCL81 and HBEC-5i cell cultures and, similar to the other validated compounds, are likely to exert antiviral activity through a molecular target in the host.

## 1. Introduction

St. Louis encephalitis virus (SLEV) is an arthropod-borne virus that may cause severe neurological pathology in humans and animals [1,2]. SLEV is a member of the *Flavivirus* genus and belongs to the Japanese encephalitis virus serocomplex, which comprises important human pathogens such as the West Nile (WNV) and Murray Valley encephalitis (MVEV) viruses [3]. SLEV is an enveloped, positive-sense single-stranded RNA virus with a non-segmented genome that encodes a single polyprotein. After being cleaved by the viral protease, the polyprotein originates three structural proteins—C, prM, and E, and seven non-structural proteins—NS1, NS2a, NS2b, NS3, NS4a, NS4b, and NS5 [4,5] involved in viral replication.

SLEV was first isolated in the 1930s in St. Louis, Missouri, during the largest outbreak of St. Louis encephalitis ever reported [6]. Since then, several outbreaks have also been reported in South America and the Caribbean [7,8]. SLEV is transmitted by Culex mosquitoes [9,10,11,12] but has also been detected in mosquitos of the Sabethes, Aedes, and Mansonia genera [12]. The transmission cycle of SLEV involves birds as a natural reservoir, whereas humans, horses, and other vertebrates are considered dead-end hosts [11,13]. 

SLEV infections are mostly asymptomatic. Symptomatic patients may present unspecific febrile illness, which is often misdiagnosed as other, more prevalent, viral diseases [14]. Patients who evolve to severe disease experience fever, headaches, drowsiness, and mental confusion, and may develop meningoencephalitis. The mortality rate ranges from 5 to 20%, with neurological sequelae affecting survivors and advanced age being the strongest risk factor identified to date [15]. There are no vaccines or specific treatments available against SLEV infection. St. Louis encephalitis is a neglected infectious disease and SLEV biology is largely unknown. 

Drug repurposing is a feasible approach based on testing compounds that have either progressed to or completed clinical tests in humans to identify candidate treatments for other diseases or applications [16]. Neglected diseases benefit immensely from drug repurposing as it requires less time and resources, and identified compounds can be made available quickly to populations at risk. Furthermore, drugs with well described targets in the literature can be used to identify host factors that could be involved in viral replication and better understand the viral pathogenesis.

We developed a phenotypic medium-throughput screening assay for SLEV infection using Vero CCL81 cells and screened the MMV COVID Box, a compound library that comprises eighty compounds with known or predicted antiviral activity against SARS-CoV-2. We identified nine hit compounds in our primary screening, of which six reduced SLEV viral loads in cell cultures. Most compounds, notably Triparanol and Fluphenazine, affected lipid distributions in treated cells, suggesting that lipidic pathways may play an important role in SLEV infection.

## 2. Materials and Methods

### 2.1. Cell Lines

Vero CCL81 cells (ATCC #CCL81), HBEC-5i, and Huh-7 were cultured in Dulbecco’s Modified Essential Medium (DMEM) (Cultilab, Campinas, Brazil) supplemented with 10% *v*/*v* fetal bovine serum (FBS) (Cultilab, Brazil) and 100 U/mL Penicillin/Streptomycin (Gibco, Grand Island, NE, USA) and maintained at 37 °C 5% CO_2_. The Huh-7 cells were provided by Dr. Marcio Chaim Bajgelman at the Brazilian Center for Research in Energy and Materials (CNPEM).

### 2.2. Compounds

The COVID Box compound library was obtained from the Medicines for Malaria Venture (MMV) (Geneva, Switzerland). Amiodarone (CAS: 1951-25-3), Imatinib (CAS: 152459-95-5), Posaconazole (CAS: 171228-49-2), and Triparanol (CAS: 78-41-1) were purchased from Sigma-Aldrich. Vidofludimus (CAS: 717824-30-1) and Fluphenazine (CAS: 69-23-8) were purchased from Cayman Chemicals. ABT-239 (CAS: 460746-46-7) was purchased from Ambeed. 7-deaza-2′-C-methyl-D-adenosine (7DMA, CAS: 443642-29-3) was synthesized at LNBio—CNPEM by Dr. Marjorie Bruder and Dr. Fabrício Naciuk. 

### 2.3. Viral Stock

The SLEV BeH 355964 strain was isolated in the late 1970s from a symptomatic female patient in Belém, Brazil, and has since been considered a laboratory-adapted strain. The viral stocks were generated by passages in Vero CCL81 cells monolayers cultivated in Dulbecco’s Modified Essential Medium (DMEM) (Cultilab, Campinas, Brazil) supplemented with 10% *v*/*v* fetal bovine serum (FBS) (Cultilab, Campinas, Brazil) and 100 U/mL Penicillin/Streptomycin (Gibco, Grand Island, NE, USA). The supernatant was collected at 48, 72, and 96 h post-infection, centrifuged, and stored at −80 °C. The stocks were quantified using viral plaque-forming assays on the Vero CCL81 cells, and the viral load was expressed as plaque-forming units (PFU)/mL of supernatant.

### 2.4. CPE-Based Medium-Throughput Screening Assay

The Vero CCL81 cells were seeded at a density of 1.7 × 10^3^; cells per well in a flat clear bottom, 384-well assay plate (Greiner Bio-One, Kremsmünster, Austria), cultured in 45 μL of DMEM, supplemented with 10% FBS (*v*/*v*), and maintained at 37 °C and 5% CO_2_. After 24 h, 1 μL of the MMV COVID Box compound library at 10 mM was added to 40 μL of DMEM in each well of an intermediate plate and, then, 15 μL was transferred to the assay plates for a final concentration of 20 μM. Cells were infected in the presence of compounds at an MOI of 0.5, by adding 15 μL of virus diluted in FBS-free media, and incubated for 72 h. The cells were stained with Hoechst-33342 after the supernatant was discarded and fixed with 4% paraformaldehyde (PFA). 

### 2.5. Confirmatory Assay

The Vero CCL81 cells were seeded at a density of 1.7 × 10^3^ cells in 45 μL per well in 384-well plates with a flat clear bottom (Greiner Bio-One, Kremsmünster, Austria) and maintained at the previously described conditions. After 24 h, 2.5 μL of hit compounds at 10 mM were added per well in an intermediate plate with 40 μL of DMEM and 10% FBS, and then 11 μL from each well was added to the cells, leading to a final concentration ranging from 100 to 0.32 μM. After the compound addition, the cells were infected with 10 µL of SLEV BeH 355964 at an MOI of 0.5 and incubated for 72 h. The cells were stained with 2 μM Hoechst-33342 after discarding the supernatant for 45 min and fixed with 4% paraformaldehyde (PFA). 

### 2.6. Viral Load Quantification

The SLEV viral load was determined by plaque assay. A 10-fold serial dilution of the samples ranging from 10^−1^ to 10^−6^ was performed and incubated with Vero CCL81 cells for 1 h for adsorption. The samples were removed, and media containing 1.5% *w*/*v* carboxymethylcellulose (Synth, Diadema, Brazil) in 2% FBS *v*/*v* DMEM was added. After 7 days of incubation at 37 °C and 5% CO_2_, the cells were fixed with 8% *w*/*v* paraformaldehyde (PFA) and washed and stained with methylene blue (Synth, Diadema, Brazil) 1% *w*/*v*. The viral load was expressed as plaque-forming units (PFU)/mL of supernatant.

### 2.7. MTT Assay

The compound toxicity to cell cultures was determined using the MTT assay. Vero cells were treated with hit compounds at different concentrations. After 48 h post-infection, the supernatant was removed and a solution containing MTT in DMEM was added and incubated for 3 h at 37 °C and 5% CO_2_. The supernatant was again removed, and the formazan crystals were solubilized in DMSO. The absorbance was measured by a microplate spectrophotometer at 490 nm and the results were normalized to non-infected/vehicle-treated controls.

### 2.8. Data Acquisition and Analysis

The medium-throughput screening and confirmatory assays’ image acquisitions were performed in Operetta High Content Image (PerkinElmer, Waltham, MA, USA), and the nuclei quantification and analysis through Columbus software 2.4.0.104236 (PerkinElmer, Waltham, MA, USA). The images were acquired from one central field per well at 10× magnification. The cytopathic effects induced by SLEV infection in the experimental groups were quantified and normalized against the negative (vehicle-tested non-infected) and positive (vehicle-tested infected) groups.

### 2.9. PDI Network Construction and BP and CC Pathway Enrichment Analysis

Clarivate’s MetaCore database (https://portal.genego.com, accessed on 17 September 2023) was used to identify the potential targets of ABT239, Amiodarone, Fluphenazine, Posaconazole, Triparanol, and Vidofludimus. The UniProt ID of the active targets was uploaded to the Cytoscape software (version 3.2.1) to construct a protein-drug interaction (PDI) network for anti-SLEV compounds. Targets shared by a minimum of two compounds were chosen for detailed analysis to elucidate the central mechanism and pathway associated with the anti-SLEV effects of ABT239, Amiodarone, Fluphenazine, Posaconazole, Triparanol, and Vidofludimus. To find the gene ontology (GO) categories (cellular component (CC), biological process (BP), and molecular function (MF)) those shared targets were evaluated in the Database for Annotation, Visualization and Integrated Discovery (DAVID; https://david.ncifcrf.gov, accessed on 18 September 2022) [17]. Their *p*-values were calculated by DAVID to assess group differences in mutually exclusive categories.

### 2.10. Statistical Analysis

The evaluation of the antiviral activity of the hits was performed using the non-parametric Kruskal–Wallis test and Dunn’s multiple comparisons test, using the Software GraphPad Prism, version 9.2. The Z′-factor value was calculated for every medium-throughput screening assay to assess quality and performance [18,19]. Z′-factor values equal to or above 0.5 were considered satisfactory.

## 3. Results

### 3.1. A Medium-Throughput Screening Pipeline for Identification of Bioactive Compounds against SLEV Infection

We developed and validated a phenotypic cell culture-based medium-throughput screening assay in the 384-well plate format to identify bioactive compounds against SLEV infection. The assay relies on the induction of cytopathic effects (CPE) by SLEV infection in Vero CCL81 cells, characterized by a reduction in the nuclei number and loss of cell monolayers. The CPE reduction was assessed through the quantification of Hoechst-stained nuclei. Compound-induced CPE inhibition was calculated relative to both the negative and positive controls, consisting of SLEV-infected or non-infected cell cultures, respectively, both treated with DMSO vehicle. Z′ factor values were calculated for each assay plate to ensure assay quality, with values over 0.5 considered valid. The MMV COVID Box library compounds were screened at final concentrations of 20 µM. Additionally, the flaviviral polymerase inhibitor 7-deaza-2′-C-methyladenosine (7DMA) was included within the set of compounds as a treatment positive control for CPE inhibition, due to its reported antiviral activity against other flaviviruses [20,21]. Cell cultures were infected with SLEV at an MOI of 0.5 shortly after the addition of compounds and incubated for 72 h. Afterward, fixed and stained cell cultures were analyzed using a high-content microscope to quantify living cells through nuclei counting. After the primary screening, compounds that inhibited SLEV-induced CPE in 40% or more were considered hits, as indicated in Figure 1.

Nine out of eighty compounds (11%) from the MMV COVID Box library showed a cytoprotective effect in our assay to different extents: Amiodarone, ABT-239, Fluphenazine, Triparanol, Imatinib, Pexidartanib, Posaconazole, Vidofludimus, and SAX-187 (Figure 2A,B). Six hit compounds were resupplied for further confirmatory assays. Despite hits on the MTS assay, Imatinib, Pexidartanib, and SAX-187 did not proceed to the validation stage. SAX-187 required a special license to purchase due to its pharmacological class, which made resupply and further validation unfeasible. Imatinib and Pexidartanib have been identified as false positive hit compounds on our previous screening campaigns using Vero CCL81 cells, and, being compounds of the same chemical class and function (tyrosine kinase inhibitors), were excluded from further testing.

Dose–response curves were generated for Vero CCL81 cultures with compound concentrations ranging from 100 µM to 0.32 µM. The half-maximal effective concentrations (EC_50_), the half-maximal cytotoxic concentrations (CC_50_), and the selectivity index (SI—CC_50_/EC_50_ ratio) were calculated and listed in Figure 2C. The compounds Posaconazole, Triparanol, ABT-239, Amiodarone, and 7DMA presented EC_50_ values in the low micromolar range (0.4 µM, 0.9 µM, 3.4 µM, 4.6 µM, and 5.1 µM, respectively). Fluphenazine and Vidofludimus were less potent, with EC_50_ values in the mid-micromolar range (10.6 µM and 8.2 µM, respectively). The compounds Posaconazole, Vidofludimus, and 7DMA were the least cytotoxic on Vero CCL81 cells. The compounds ABT-239, Posaconazole, Triparanol, Vidofludimus, and 7DMA were also selective, with SI values of 12 or greater.

### 3.2. Hit Validation

To determine whether hit compounds had antiviral or solely cytoprotective properties against SLEV, we performed an orthogonal assay to test hit compounds at 20 µM in three different cell lines infected with SLEV at an MOI of 0.1. The compound 7DMA was included as a positive-control treatment for reduction in viral replication, at the concentration of 40 µM. Cell viability was assessed using the MTT assay, and infectious viral load was quantified in the supernatant using plaque-forming assays. 

Of all the compounds tested on Vero CCL81, Posaconazole was the most toxic at 20 µM, reducing cell viability by 40% on average when compared to non-treated controls (Figure 3A). The other compounds, including 7DMA, showed limited (less than 10%) or no reduction in cell culture viability, indicating that these compounds were not toxic to the cell culture. 

In terms of antiviral activity, Triparanol was the most potent compound against SLEV infection on Vero CCL81, reducing the viral load by over 10,000-fold relative to vehicle-treated controls (Figure 3B,C). Fluphenazine also showed significant antiviral activity, reducing the viral load up to 1000-fold at 20 µM. The compounds ABT-239, Amiodarone, and Vidofludimus also reduced viral loads approximately 10-fold, which was comparable to the antiviral activity promoted by treatment with the control 7DMA. As indicated in Figure 3C, the antiviral effects of the most potent compounds in Vero CCL81 cells, Triparanol and Fluphenazine, were observed through a reduction in the total number of viral lysis plaques, and not on lysis plaque shape or radius.

We also tested the seven hit compounds and the control 7DMA on the human cell lines HBEC-5i and Huh-7. Notably, Fluphenazine and Triparanol were highly toxic to Huh-7 cells, leading to a 90% loss in cell viability when compared to vehicle-treated cells (Figure 4A). The Posaconazole reduced cell culture viability by 50%, while Amiodarone, ABT-239, Vidofludimus, and 7DMA were less toxic, with reductions in cell viability of approx. 10% in comparison to vehicle-treated controls. Only the 7DMA treatment reduced viral load (10-fold) in SLEV-infected Huh-7 cell cultures without a significant reduction in cell culture viability (Figure 4B). The Amiodarone treatment reduced viral loads in Huh-7 culture supernatants without cell toxicity, but the antiviral effect was not significant in comparison to the infected controls.

The treatments with ABT-239, Amiodarone, and Fluphenazine were not toxic on HBEC-5i cells (Figure 4C). However, the Posaconazole and Triparanol reduced cell viability by approximately 70% and 60%, respectively. The cell culture viability after the Vidofludimus treatment was up to 90% in comparison to the vehicle-treated non-infected group. Regarding antiviral activity, the Amiodarone and Fluphenazine promoted over 10-fold and 100-fold reductions in viral titers in culture supernatant, respectively (Figure 4D). No antiviral activity was observed after the Vidofludimus treatment.

Given the pronounced antiviral activity of Triparanol and Fluphenazine on Vero CCL81, we tested both compounds at lower concentrations ranging from 20 µM to 2.5 µM on HBEC-5i cells at MOI 0.1 to evaluate whether they have any antiviral activity at non-toxic concentrations, i.e., were selective in HBEC-5i. Triparanol was not toxic at lower concentrations and significantly (*p*  <  0.05) reduced the viral load at the concentration of 10 µM, over 10-fold compared to the vehicle-treated infected group (Figure 5A,B). The antiviral effect of Triparanol was subsequently lost at concentrations of 5 µM and 2.5 µM. Although Fluphenazine is not toxic at any of the concentrations tested on HBEC-5i cells, its antiviral effect was only pronounced at 20 µM, with an over 50-fold reduction in viral load (Figure 5C,D).

### 3.3. Identification of Possible Host-Targeted Mechanisms of Action for Compounds with Antiviral Activity against SLEV 

ABT-239, Amiodarone, Fluphenazine, Posaconazole, Triparanol, and Vidofludimus were developed for the treatment of various human conditions, which do not include viral infections. We investigated which mechanisms of action and molecular targets have been identified and characterized for our six validated compounds in search of possible mechanisms by which these compounds would exert antiviral activity against SLEV. An analysis performed using the Metacore database led us to create an interaction network that indicated multiple targets and pathways in the human host (Figure 6). Each compound possesses distinct targets, yet there is substantial overlap in the targets shared by multiple compounds. Amiodarone is the compound with the highest amount of information in the database, facilitating the identification of multiple unique targets as well as several shared targets with Fluphenazine, ABT-239, Triparanol, and Posaconazole, either with all four compounds or with compound pairs (e.g., Posaconazole/Fluphenazine or ABT-239/Fluphenazine). The majority of known targets for ABT239 are also targeted by some of the other compounds assessed in this study, marking them as interesting targets in SLEV infection. However, the limited information available in the Metacore database regarding Triparanol and Vidofludimus somewhat restricts the analysis of these compounds. The shared targets identified among the anti-SLEV compounds prominently include protein families such as histamine receptors (HRH), Cytochrome P450 (CYP2), and ATP-binding cassette transporter (ABCB). With the exception of Vidofludimus and Triparanol, all analyzed compounds target one or more proteins within these families. Triparanol shares a single molecular target with Amiodarone, and Vidofludimus was the only compound without molecular targets shared with other anti-SLEV compounds. Insufficient data in the Metacore database hampers comprehensive analysis of molecular processes and mechanism of action for the two compounds.

We selected the shared molecular targets (Figure 6A, red circles) and ranked the associated biological processes (Figure 6B) and cellular compartments (Figure 6C). Our analysis indicated that pathways related to xenobiotic metabolic and catabolic processes may be significantly affected, which are frequently altered in the context of pharmacological treatments. Importantly, the molecular targets of anti-SLEV compounds also converge on host pathways involved in lipid and steroidal metabolism (Figure 6B). The cellular compartments mostly affected by those compounds include the endoplasmic reticulum, cellular membranes, and associated components (Figure 6C). Overall, our analysis suggests that molecular targets and biological processes altered by our six validated compounds are biased towards the modulation of the host lipidic metabolism and associated with changes in lipid membranes and associated structures, such as transmembrane receptors and endosomes. 

## 4. Discussion

The emergence and reemergence of mosquito-borne viral diseases have caused a significant impact on human health worldwide in the past decades [22,23]. The lack of specific treatments for the majority, if not all, mosquito-borne viral diseases, combined with inefficient prophylaxis, either by controlling the mosquito vector or by vaccination, and the ability of these viral pathogens such as SLEV to cause severe disease highlights the urgency of this situation. Hence, we developed and validated a phenotypic cell culture-based medium-throughput screening assay to identify antiviral compounds against SLEV infection in the MMV COVID Box library, a compound library biased towards drug repurposing. To our knowledge, we have presented the first compound screening campaign against SLEV.

The screening of compound libraries populated with compounds approved for human use, aimed at drug repurposing, is an interesting strategy for the development of antiviral therapies against neglected infectious diseases. As the safety profile of the repurposed drugs are already known and well-described in the literature, this approach requires less time and resources in comparison to the discovery of novel compounds. Therefore, it could fast-track the identification of new therapies and become quickly available to the population. Furthermore, this approach can also be useful to elucidate viral biology, as the well-known targets could suggest important host factors and metabolic pathways in SLEV infection.

Our assay disclosed hit compounds that were later confirmed in a different human cell line. Despite this cell culture-based screening assay being an indirect method to screen antiviral drugs, it is based on the direct association between viral replication and virus-induced CPE by flaviviruses in Vero CCL81 and other permissive cell lines that are commonly used in arboviral research. Furthermore, this assay can be adapted for high-throughput screening campaigns with larger compound libraries. This compound screening assay was already developed in a 384-well plate format and the Z′-factor values were calculated for each assay plate independently, allowing the use of as many assay plates and technical replicates as necessary.

The pharmacological classes of selected hit compounds are diverse, including antilipemic, antiarrhythmic, immunomodulatory, antipsychotic, and antitumor agents, suggesting the existence of multiple molecular targets/pathways that can be modulated for an antiviral effect against SLEV. Cellular metabolism and permissiveness to SLEV infection can vary greatly among different cell types and tissues [24,25], which motivated the validation of hit compounds on three distinct cell lines. In addition to Vero CCL81 cells, which are derived from an African green monkey kidney and highly permissive to SLEV, compound activity was evaluated in (1) human brain microvascular endothelial cells (HBEC-5i), due to the neuroinvasiveness associated with SLEV in patients developing severe neurological disease; and in (2) human hepatoma cell line Huh-7, which is permissive to infection with multiple arboviruses and represents a frequently targeted organ in systemic arboviral disease. As expected, our data showed that compounds had distinct antiviral activity potencies and toxicity in different cell types. Triparanol was the most potent hit compound in the screening and validation assays, reducing the viral load 10,000-fold in Vero CCL81 cells at 20 µM. Under the same conditions, Triparanol was less potent and less selective in HBEC-5i cells, and highly toxic to Huh-7. Triparanol is an antilipemic that inhibits 24-dehydrocholesterol reductase (DHCR24)—an enzyme that catalyzes the conversion of desmosterol into cholesterol [26]. Cholesterol and sphingolipids are essential for HCV maturation and infectivity [27], and cholesterol has been demonstrated to be important during the infection and replication of several flaviviruses [28,29,30]. Another study suggested that DHCR24 expression increases during viral infection and plays an important role during HCV replication [31]. 

Fluphenazine is a first-generation antipsychotic used in the treatment of schizophrenia by acting as a dopamine D2 receptor antagonist [32]. Fluphenazine is also a functional inhibitor of acid sphingomyelinase (FIASMA) [33], a class of compounds that inhibit acid sphingomyelinase (ASM), an enzyme that cleaves sphingomyelin into phosphorylcholine and ceramide [34], mainly within lysosomes. Previous studies have shown that ceramide plays an important role in bacterial [35] and arboviral infections, such as in the entry and egress of JEV [36]. The inhibition of ASM leads to a lower concentration of ceramide in cells, and thus, treatment with FIASMAs might also impair SLEV infection. Indeed, our data showed that Fluphenazine reduced over 1000- and 100-fold the viral load in Vero CCL81 and HBEC-5i cells, respectively. This reduction might be associated with the viral entry process since ceramide modulates endocytosis [37] and SLEV, like other flaviviruses, enter cells through a clathrin-mediated endocytosis process [38,39]. FIASMAs also appear to have good permeability across the blood−brain barrier [40], which can be particularly interesting in therapeutical approaches for neuroinvasive viruses such as SLEV.

Triparanol and Fluphenazine are also described as cationic amphiphilic compounds (CAD) [41]. CADs are molecules with a hydrophilic amine that can be protonated, and a hydrophobic tail composed of an aromatic or aliphatic ring capable of anchoring in cell membranes or the lipid bilayer [42]. They can accumulate in acidic intracellular compartments such as late endosomes/lysosomes (LE/Lys) because when the hydrophilic amine becomes protonated due to an acidic environment, the molecule can no longer pass through the membrane [43]. This phenomenon leads to drug accumulation inside these compartments and promotes drug-induced phospholipidosis (DIPL), which occurs when phospholipids accumulate in cells or tissues [42]. The exact mechanism of why DIPL affects viral infection remains unclear, but it may involve dysregulation in host cell lipidic homeostasis [41].

Among the hits with low or moderate activity, we found Amiodarone, a broad-spectrum class III antiarrhythmic that acts as a vasodilator [43]. The compound acts by blocking potassium channels, but it can also block calcium channels [44] and interact with alpha- and beta-adrenergic receptors [43]. Furthermore, Amiodarone is also characterized as a cationic amphiphilic compound (CAD) [41] and FIASMA [45]. Recent studies have shown inhibition of infection by SARS-CoV-2 after treatment with Amiodarone [44], reinforcing the hypothesis that the deregulation of lipid homeostasis in the host has a negative impact on infection. Vidofludimus is an immunomodulatory compound developed for the treatment of multiple sclerosis, inflammatory diseases of the digestive tract, and other chronic and autoimmune diseases [46]. Vidofludimus inhibits the dihydroorotate dehydrogenase (DHODH), an enzyme that catalyzes the oxidation of dihydroorotate to orotate in the de novo synthesis of pyrimidine [47,48]. Inhibition of DHODH leads to the depletion of pyrimidine nucleosides, which affects the replication of RNA viruses [49,50,51]. ABT-239 reduced SLEV replication by approx. 10-fold on both Vero CCL81 and HBEC-5i cells. ABT-239 is a histamine H3 receptor antagonist [52] and transient vanilloid receptor type 1 (TRPV1) antagonist proposed for the treatment of schizophrenia and attention-deficit/hyperactivity disorder (ADHD) [53]. TRPV1 antagonists exacerbated the oxLDL-induced lipid accumulation in macrophages [54], suggesting that TRPV1 might have a role in lipidic metabolism. Also, it was described that an antagonist of TRPV1 decreased CHIKV infection in macrophages [55].

Despite being a hit on screening assay, Posaconazole substantially reduced the cell viability of all cell types tested. Posaconazole is an antifungal that inhibits 14-α-demethylase (CYP51), an enzyme that acts in the steroid biosynthesis pathway [56]. Previous studies have shown it was able to inhibit DENV replication in vitro through its interaction with oxysterol-binding protein (OSBP), a lipid-transport transmembrane protein responsible for transporting sterol and phosphatidylinositol 4-phosphate (PI4P) between the endoplasmic reticulum and Golgi complex [57] and necessary for the replication of several types of viruses [58]. Also, in combination with two other drugs, Posaconazole blocked the entry of EBOV by acid sphingomyelinase inhibition, the release of lysosomal calcium, and NPC1, a membrane protein essential for mediating intracellular cholesterol trafficking in mammals [59]. Furthermore, it had antiviral activity in the early stages of infection against Parechovirus A3 (PeV-A3) [60].

In summary, our results showed that a cell culture-based screening assay led to the identification of antiviral compounds against SLEV in a compound library set for drug repurposing. Such an approach is a feasible strategy to accelerate the discovery of candidate treatments for neglected arboviral diseases such as St. Louis encephalitis. Our data suggest that the identified hit compounds converge to targets in host lipidic pathways, notably for Triparanol, which had greater antiviral activity in our validation assays. Changes in lipidic homeostasis might impair SLEV infection and corroborate previous findings indicating an important role of lipidic metabolism in Flavivirus infection, highlighting the need for a more complete understanding of SLEV biology.

## Figures and Tables

**Figure 1 viruses-15-02416-f001:**
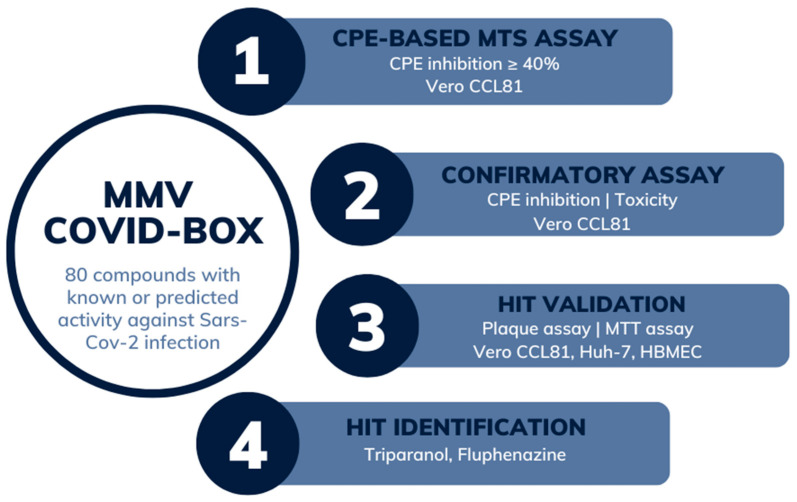
Medium-throughput screening pipeline to identify antiviral compounds against St. Louis encephalitis virus infection. (1) The MMV COVID Box library, which comprises eighty compounds with known or predicted activity against SARS-CoV-2, was screened at 20 µM using a phenotypical assay of SLEV infection in cell culture. Briefly, hit compounds were selected based on their ability to reduce the CPE induced by SLEV infection in Vero CCL81 cells, which was quantified by a high-content fluorescence microscope detecting Hoechst-stained nuclei in cell cultures. Nine hit compounds were selected due to inhibition of at least 40% of the CPE induced by viral infection at 72 h post-infection. (2) Six compounds were resupplied for the confirmatory assay. Dose–response curves were created for Vero CCL81 cell cultures for the calculation of EC_50_ and CC_50_ values. (3) Hit validation was performed using Vero and the human cell lines HBEC-5i and HUH-7. The antiviral activity was confirmed by plaque-forming assays. (4) The pipeline led to the identification of Triparanol and Fluphenazine with antiviral activity against SLEV infection.

**Figure 2 viruses-15-02416-f002:**
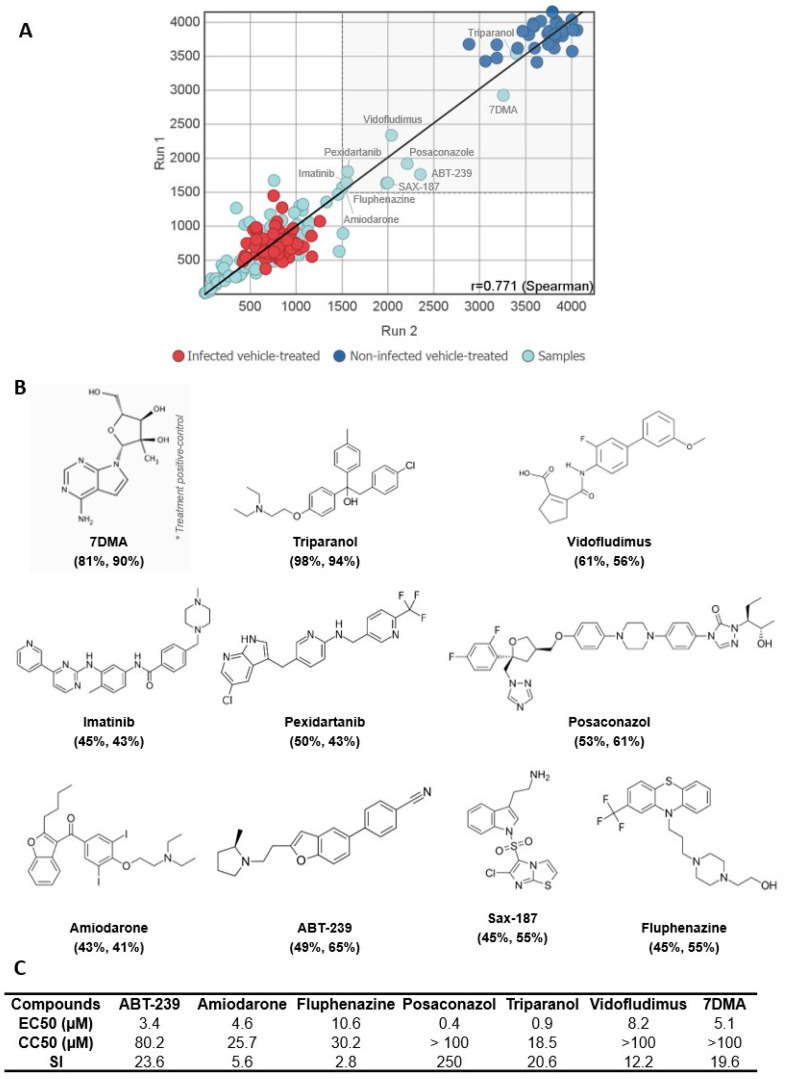
Identification of cytoprotective compounds against SLEV infection in the MMV COVID Box. (**A**) Correlation of two screening runs of the compound library at 20 µM for the ability to reduce the CPE induced by SLEV infection in Vero cells after 72 h. The positive (non-infected cells treated with vehicle) and negative (SLEV-infected cells treated with vehicle) groups are represented by dark blue and red, respectively. 7DMA was used as a positive control for treatment and tested at 40 µM. Of the eighty compounds screened, nine inhibited at least 40% of CPE and were classified as hit compounds. The cut-off and range adopted to select the hit compounds are highlighted in gray. (**B**) Chemical structure of hit compounds from the MMV COVID Box and the treatment positive control 7DMA. * 7-deaza-2′-C-methyladenosine (7DMA) was included within the set of compounds as a treatment positive control for CPE inhibition, due to its reported antiviral activity against other flaviviruses. (**C**) EC_50_, CC_50_, and Selectivity Index (SI) values for confirmed hit compounds, including 7DMA as a positive control. Values of EC_50_/CC_50_ were calculated from technical replicates of the dose–response curve ranging from 100 µM to 0.32 µM of drug concentration.

**Figure 3 viruses-15-02416-f003:**
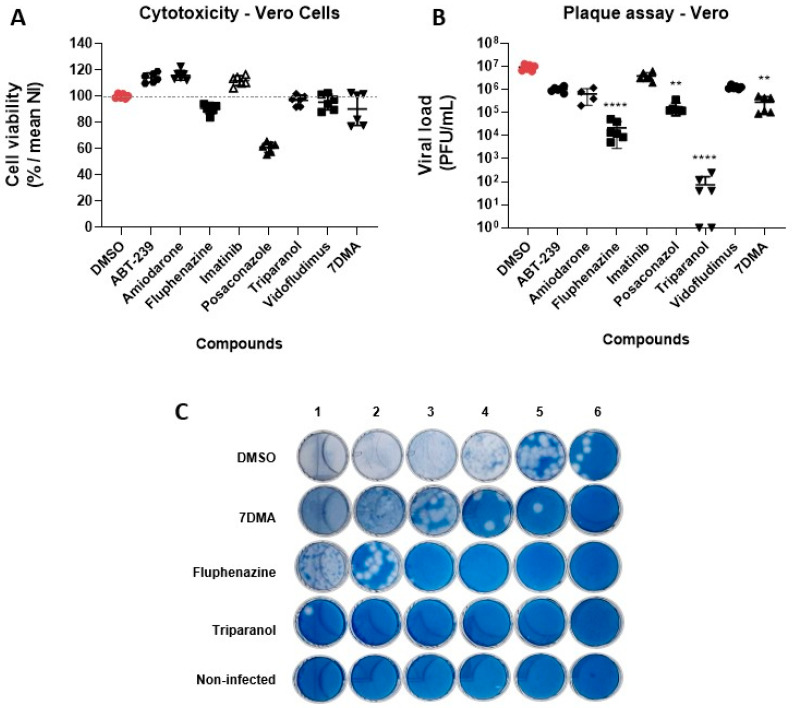
Hit compound validation of antiviral activity against SLEV in Vero CCL81 cells. (**A**) Compound cytotoxicity was assessed through MTT assay. (**B**) Viral load measured by plaque assay. (**C**) Plaque assays of the most promising hits in Vero cells. A serial dilution ranging from 10^−1^ to 10^−6^ was performed on each sample. The antiviral activity of compounds promoted the reduction in lysis plates in comparison to positive control (vehicle-treated group). ** *p* < 0.01, **** *p* < 0.0001 relative to the virus-infected, vehicle-treated control group. Data represent two independent experiments (*n* = 6).

**Figure 4 viruses-15-02416-f004:**
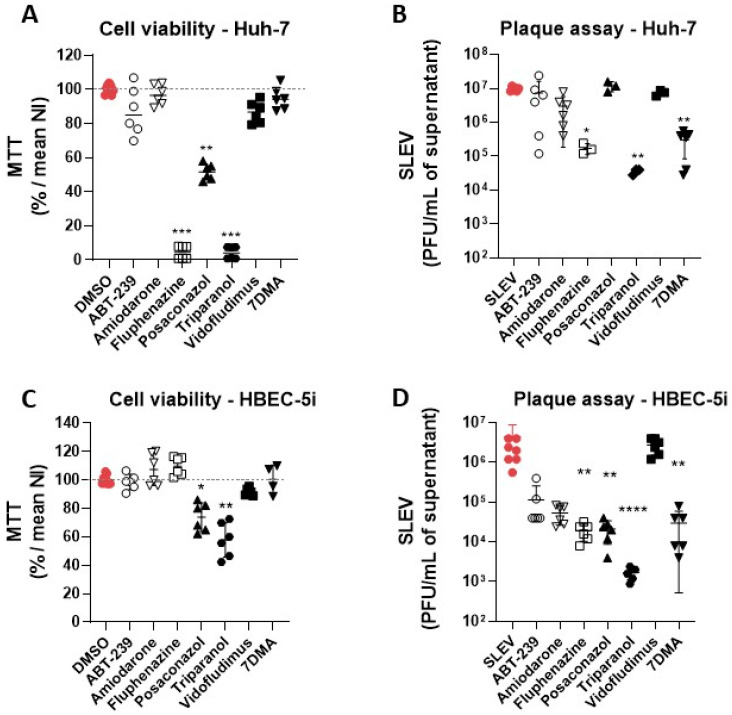
Hit compound validation of antiviral activity against SLEV in Huh-7 and HBEC-5i cells. Cytotoxicity and antiviral activity of hit compounds were measured by MTT assay and plaque assay in (**A**,**B**) Huh-7 and (**C**,**D**) HBEC-5i cell lines, respectively. The experiments were performed after 48 h of infection at MOI of 0.1. * *p* < 0.05, ** *p* < 0.01, *** *p* < 0.001, **** *p* < 0.0001 relative to the virus-infected, vehicle-treated control group. Data represent two independent experiments (*n* = 6).

**Figure 5 viruses-15-02416-f005:**
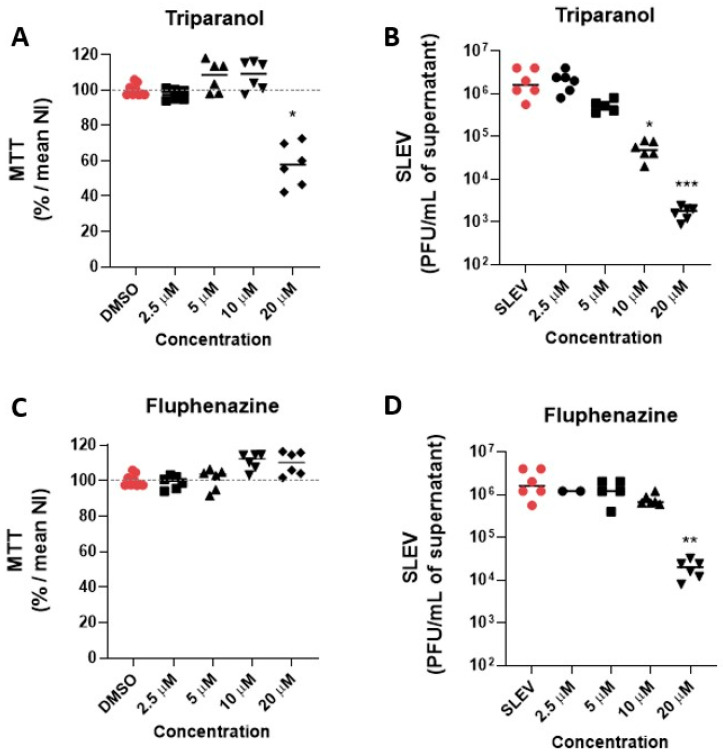
Triparanol and Fluphenazine have antiviral activity against SLEV infection in HBEC-5i cells. Cytotoxicity and antiviral activity of Triparanol (**A**,**B**) and Fluphenazine (**C**,**D**) at lower concentrations ranging from 20 µM to 2.5 µM were measured by MTT assay and plaque assay in HBEC-5i cells. The experiments were performed after 48 h of infection at MOI of 0.1. * *p*  <  0.05, ** *p*  <  0.01, *** *p*  <  0.001 relative to the virus-infected, vehicle-treated control group. Data represent two independent experiments (*n* = 6).

**Figure 6 viruses-15-02416-f006:**
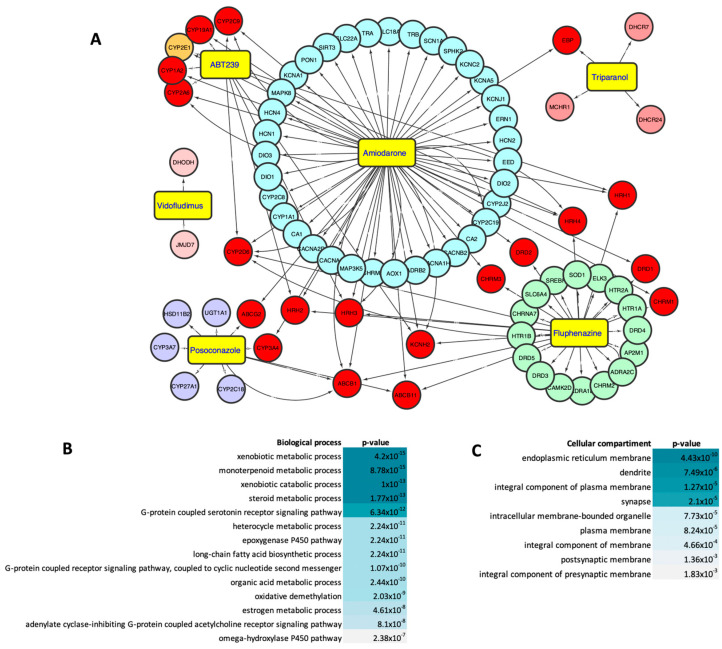
Interaction network of compounds with validated antiviral activity against SLEV. (**A**) Circle-shaped nodes indicate reported protein targets of ABT-239, Amiodarone, Fluphenazine, Posaconazole, Triparanol, and Vidofludimus according to Metacore. Targets were manually categorized and color-coded based on their association with respective drugs. Red nodes represent shared targets for two or more compounds. (**B**) Biological process enrichment analysis of overlapped targets. (**C**) Cellular compartment enrichment analysis of overlapped targets.

## Data Availability

The data presented in this study are available on request from the corresponding author.

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
