# Peer review of "Phenotypical Screening of an MMV Open Box Library and Identification of Compounds with Antiviral Activity against St. Louis Encephalitis Virus"

_viruses, 2023, doi:10.3390/v15122416_

Round 1
Reviewer 1 Report
Comments and Suggestions for Authors
The authors ran a phenotypic screen of 80 known compounds against St Louis encephalitis virus in Vero CCL81 cells using a CPE-based readout. Six of the nine hits initially identified were validated through plaque assay and cytotoxicity assay, and their antiviral activity and cytotoxicity were further evaluated in two human cells lines. The authors then focused on two of the drugs, Triparanol and Fluphenazine, looking by microscopy at the cholesterol distribution in infected and non-infected conditions, and finally analyzed and clustered the six drugs according to their know targets and biological processes targeted.
The screening approach is novel for this virus and the study represents an interesting proof of concept. The narrative flows from figure 1 to figure 5, but in the second part of the manuscript the overall message results not on focus.
The following points could improve the manuscript.
Major comments:
1) The interest of chemical compound screening using known drugs can apply to two areas: i) the identification of compounds with known mechanism of action and safety profile that could be repurposed and could fast-track the identification of new antiviral treatments; ii) the use of drugs with known mechanisms of action to focus on the biology of the virus and the use of the hits to indirectly identify host factors that could be involved in viral replication
The current manuscript fails at clarifying these two aspects, touching a little bit both areas, without providing enough depth or context for either of them. The manuscript could benefit from a clearer narrative, by clarifying the following:
a. The study presents the successful optimization of an assay that could be used for future screen of antiviral candidates. This specific set unfortunately doesn’t provide direct drug repurposing opportunities, because of lack of reconfirmation of antiviral activity in human cell lines and/or limited selectivity across different models, and this should be mentioned.
HOWEVER
b. Although the hits are not promising candidates for drug repurposing, these compounds can still bring value to the understanding of the biology of the virus, having known mechanism of action/targets.
In support of this, a time-of-drug addition kinetics ideally run in a condition of single-cycle of infection will help to identify the specific steps of the viral replication inhibited by each compound and differentiate entry inhibitors, inhibitors of replication, and late-stage inhibitors. This will provide more information to speculate about potential mechanism of action/biological processes involved in SLEV biology.
2) The interpretation of plaque assay data at concentrations where a compound shows some levels of cytotoxicity, incorporates a bias in data interpretation. Considering the compounds have different EC50 and selectivity values in Vero CCL81 cells, selecting a same fixed concentration for all the drugs tested in this assay is a limiting factor (especially in different cell lines where no tox range for the drug has been assessed). Differently from a primary screen, where a standardized single concentration can be selected for all the compounds, considering the number of compounds to test, validation assays should take into consideration the limitations of this approach and avoid those limitations. At least for the drugs showing cytotoxicity at 20 uM, a DR curve to evaluate cytotox and CC50 in each cell line should be run first (either MTT or DAPI counting will be acceptable). This will help to select for each compound at least a concentration at which the drug is not cytotoxic, to then run a plaque assay and more reliably evaluate the antiviral activity, removing the bias of toxicity-driven inhibition. Ideally, each compound should also be tested in dose-response through an antiviral assay in Huh-7 and HBMEC cells, to allow calculate the EC50 values (either by CPE assay, IF, plaque assay or any alternative antiviral readout) and corresponding selectivity.
3) Figure 6 doesn’t strengthen the message of the study and doesn’t add any knowledge to the manuscript and the field: i) The cells are not labeled for infection markers, thus it is not possible to differentiate infected vs non-infected cells in the SLEV condition and/or to evaluate eventual co-localizations of cholesterol with viral proteins; ii) the effect of Triparanol on cholesterol is known, and no real correlation of cholesterol distribution and viral infection is presented here or investigated further. i.e. If cholesterol is important, one would expect U18666A to also have an antiviral effect on SLEV, but this has not been explored in the study. The direct implication of cholesterol metabolism on SLEV infection is not addressed. Although cholesterol very likely has a key role during SLEV replication (as well documented for other flaviviruses and RNA viruses), currently this figure neither answers that question nor enriches the manuscript. What message are the authors trying to deliver with this figure? If the authors want to prove the importance of cholesterol on SLEV replication, SEVERAL additional experiments would be required, and the overall focus of the manuscript would change. I suggest removing figure 6 from the manuscript.
Minor comments:
1) Figure 2: Since the nine hits only show partial inhibition at 20 uM, it would be helpful to include in the manuscript the dose-response curves (for antiviral activity and cytotoxicity in uninfected cells) for the compounds listed in Figure 2C. This will better clarify the efficacy of these compounds and their potential as therapeutics (i.e. are all of the compounds reaching 100% inhibition at 100 uM, or only partial efficacy is reached even at the highest concentration?).
2) In Materials and Methods, the protocol for the confirmatory assay is well detailed, while the protocol for the CPE-based medium-throughput screening assay misses some information (i.e. volumes of media used for intermediate dilution, volumes of virus added). Please, provide the missing information. Also, please clarify how the CPE level was quantified: though high-content based quantification based on DAPI staining as indicated in lines 177-289 and suggested in M&M, or through MTS assay as indicated in Figure 1 “CPE-BASED MTS ASSAY”?
3) In line 173-175 is indicated that 7DMA was included as a positive control for CPE inhibition. It is not clear if this compound was included in all the plates as a real positive control for screening purposes, or just included within the set of compounds tested.
For antiviral screening purposes, whenever available, it would be more appropriate to calculate the Z’ factor using an antiviral with known activity as positive control rather than non-infected cells, being more representative of the dynamic range of inhibition achievable at the specific MOI. If 7DMA was included in all the plates, please provide Z’ factor using it as positive control. If it was only included within the set of compounds, considering it wasn’t previously reported to inhibit SLEV, the sentence shouldn’t be included, but rather mentioned in lines 191-193 that one of the hits was 7DMA, with previously reported anti-flavivirus activity.
4) The rationale why Imatinib, 194 SAX-187, and Pexidartanib were not resupplied is unclear. Please, clarify why the other 6 hits were prioritized.
Comments on the Quality of English Language
Line 113: “and a media” remove “a”
Line 333 misspelling: endosomes
Author Response
The authors ran a phenotypic screen of 80 known compounds against St Louis encephalitis virus in Vero CCL81 cells using a CPE-based readout. Six of the nine hits initially identified were validated through plaque assay and cytotoxicity assay, and their antiviral activity and cytotoxicity were further evaluated in two human cells lines. The authors then focused on two of the drugs, Triparanol and Fluphenazine, looking by microscopy at the cholesterol distribution in infected and non-infected conditions, and finally analyzed and clustered the six drugs according to their know targets and biological processes targeted.
The screening approach is novel for this virus and the study represents an interesting proof of concept. The narrative flows from figure 1 to figure 5, but in the second part of the manuscript the overall message results not on focus.
The following points could improve the manuscript.
» Major comments:
- The interest of chemical compound screening using known drugs can apply to two areas:
- the identification of compounds with known mechanism of action and safety profile that could be repurposed and could fast-track the identification of new antiviral treatments;
- the use of drugs with known mechanisms of action to focus on the biology of the virus and the use of the hits to indirectly identify host factors that could be involved in viral replication
The current manuscript fails at clarifying these two aspects, touching a little bit both areas, without providing enough depth or context for either of them. The manuscript could benefit from a clearer narrative, by clarifying the following:
- The study presents the successful optimization of an assay that could be used for future screen of antiviral candidates. This specific set unfortunately doesn’t provide direct drug repurposing opportunities, because of lack of reconfirmation of antiviral activity in human cell lines and/or limited selectivity across different models, and this should be mentioned.
Response: Thank you for your suggestion. Indeed, our results are far from supporting the immediate drug repurpose for clinical trials in St. Louis encephalitis patients. However, this is the first study indicating that compounds with antiviral activity can be found within available FDA-approved compounds or those that advanced through different stages of drug development. We validated the antiviral activity of our hit compounds on human cell lines HBEC-5i and Huh-7, as shown in figure 4 of this manuscript. Besides, we realize this topic should be better explored in our manuscript and included a new paragraph (lines 344 to 352) in our discussion, addressing the advantages and limitations of the drug repurposing for a disease such as St. Louis encephalitis, in accordance with your assessment.
HOWEVER
- Although the hits are not promising candidates for drug repurposing, these compounds can still bring value to the understanding of the biology of the virus, having known mechanism of action/targets.
In support of this, a time-of-drug addition kinetics ideally run in a condition of single-cycle of infection will help to identify the specific steps of the viral replication inhibited by each compound and differentiate entry inhibitors, inhibitors of replication, and late-stage inhibitors. This will provide more information to speculate about potential mechanism of action/biological processes involved in SLEV biology.
Response: We appreciate and agree with your insights, as a time-of-addition assay would provide key information about the mechanism of action of our hit compounds. However, search of molecular mechanisms for our hit compounds, though critical for drug development, are outside the scope of this paper. We focused on the development of a MTS assay, on the identification and validation of hit compounds that present antiviral activity against SLEV in cell culture. We believe that the insights provided by our manuscript are first steps to the discovery of bioactive molecules and potential pathways to be targeted in SLEV infection.
- The interpretation of plaque assay data at concentrations where a compound shows some level of cytotoxicity, incorporates a bias in data interpretation. Considering the compounds have different EC50 and selectivity values in Vero CCL81 cells, selecting a same fixed concentration for all the drugs tested in this assay is a limiting factor (especially in different cell lines where no tox range for the drug has been assessed). Differently from a primary screen, where a standardized single concentration can be selected for all the compounds, considering the number of compounds to test, validation assays should take into consideration the limitations of this approach and avoid those limitations. At least for the drugs showing cytotoxicity at 20 uM, a DR curve to evaluate cytotoxicity and CC50 in each cell line should be run first (either MTT or DAPI counting will be acceptable). This will help to select for each compound at least a concentration at which the drug is not cytotoxic, to then run a plaque assay and more reliably evaluate the antiviral activity, removing the bias of toxicity-driven inhibition. Ideally, each compound should also be tested in dose-response through an antiviral assay in Huh-7 and HBMEC cells, to allow calculate the EC50 values (either by CPE assay, IF, plaque assay or any alternative antiviral readout) and corresponding selectivity.
Response: We appreciate the considerations from reviewer 2. The fixed concentration for testing of antiviral activity shown on Figures 3 and 4 was selected based on the results shown in Figure 1. The EC50 and CC50 values calculated for Vero CCL81 cells indicated that 20 µM was the concentration in which we could observe antiviral activity with minimal toxicity for most compounds. As indicated in the following figures, EC50 and CC50 values vary greatly between different cell types, so instead of performing DR curves for every compound in every cell type, which would be little informative and time-consuming, we focused on performing DR curves (including both MTT and plaque assay) for the best compounds: Fluphenazine and Triparanol. As shown, these compounds present antiviral activity, reducing infectious viral load in cell culture supernatant, in concentrations that are not toxic to HBEC-5i cells, though with reduced selectivity.
- Figure 6 doesn’t strengthen the message of the study and doesn’t add any knowledge to the manuscript and the field:
- The cells are not labeled for infection markers, thus it is not possible to differentiate infected vs non-infected cells in the SLEV condition and/or to evaluate eventual co-localizations of cholesterol with viral proteins;
- the effect of Triparanol on cholesterol is known, and no real correlation of cholesterol distribution and viral infection is presented here or investigated further. i.e. If cholesterol is important, one would expect U18666A to also have an antiviral effect on SLEV, but this has not been explored in the study. The direct implication of cholesterol metabolism on SLEV infection is not addressed. Although cholesterol very likely has a key role during SLEV replication (as well documented for other flaviviruses and RNA viruses), currently this figure neither answers that question nor enriches the manuscript. What message are the authors trying to deliver with this figure? If the authors want to prove the importance of cholesterol on SLEV replication, SEVERAL additional experiments would be required, and the overall focus of the manuscript would change. I suggest removing figure 6 from the manuscript.
Response: We have reflected this comment and decided to remove figure 6 and associated text from the manuscript, as we realized from the reviewers comments that data was incomplete and added little to the manuscript.
Minor comments:
- Figure 2:  Since the nine hits only show partial inhibition at 20 uM, it would be helpful to include in the manuscript the dose-response curves (for antiviral activity and cytotoxicity in uninfected cells) for the compounds listed in Figure 2C. This will better clarify the efficacy of these compounds and their potential as therapeutics (i.e. are all of the compounds reaching 100% inhibition at 100 uM, or only partial efficacy is reached even at the highest concentration?).
Response: We included the EC50, CC50 and SI values for the hit compounds in a table for conciseness, and to present information in a format frequently observed in similar papers. Our phenotypical screening assay is indirect, i.e., the assessment of CPE inhibition is an indirect assessment of antiviral activity, and potentially false-positive. Thus, we focused on presenting dose-response curves of the most promising hits compounds (Figure 5), Triparanol and Fluphenazine, for which we confirmed antiviral activity in at least two cell types tested.
- In Materials and Methods, the protocol for the confirmatory assay is well detailed, while the protocol for the CPE-based medium-throughput screening assay misses some information (i.e. volumes of media used for intermediate dilution, volumes of virus added). Please, provide the missing information. Also, please clarify how the CPE level was quantified: though high-content based quantification based on DAPI staining as indicated in lines 177-289 and suggested in M&M, or through MTS assay as indicated in Figure 1 “CPE-BASED MTS ASSAY”?
Response: Based on your suggestion, we revised section 3.1 and rewrote the paragraph to clarify that CPE reduction was assessed based on the quantification of Hoechst-stained cell nuclei. We also included this description on the legend of figure 1. We hope the changes provide a more precise description of our assay. Additionally, we included the missing parameters adopted on MTS assay (Materials and Methods, section 2.4).
- In line 173-175 is indicated that 7DMA was included as a positive control for CPE inhibition. It is not clear if this compound was included in all the plates as a real positive control for screening purposes, or just included within the set of compounds tested. For antiviral screening purposes, whenever available, it would be more appropriate to calculate the Z’ factor using an antiviral with known activity as positive control rather than non-infected cells, being more representative of the dynamic range of inhibition achievable at the specific MOI. If 7DMA was included in all the plates, please provide Z’ factor using it as positive control. If it was only included within the set of compounds, considering it wasn’t previously reported to inhibit SLEV, the sentence shouldn’t be included, but rather mentioned in lines 191-193 that one of the hits was 7DMA, with previously reported anti-flavivirus activity.
Response: This is a valid assessment for the design of an MTS assay. 7DMA was not used to calculate the Z'-factor because 7DMA is not an antiviral treatment for SLEV infection, but rather an antiviral compound with known activity against a range of flaviviruses. We used vehicle-treated infected cells as negative and vehicle-treated non-infected cells as positive controls to calculate the Z' value because they increase assay range, allowing us to assess different levels of compound efficacy (moderate hits, potent hits) and to establish higher Z' values in our quality control. Hence, we modified the text to clarify that we 7DMA was included as a treatment positive control for treatment (Section 3.1 lines #173 and #175). We also modified the color of 7DMA on the Figure 2B to emphasize it is a control treatment that we added to the library.
- The rationale why Imatinib, 194 SAX-187, and Pexidartanib were not resupplied is unclear. Please, clarify why the other 6 hits were prioritized.
Response: Indeed, compounds Imatinib, Pexidartanib and SAX-187 are missing in the selection of compounds for confirmation and validation. We incorporated a new paragraph on section 3.2 (lines #192 to #198) explaining that SAX-187 required a special license for purchase, which made resupply and further testing unfeasible. Imatinib and Pexidartanib have been identified as false positive hit compounds on our previous screening campaigns using Vero CCL81 cells, and being compounds of the same chemical class and function (tyrosine kinase inhibitors), were excluded from further analysis.
Comments on the Quality of English Language:
Line 113: “and a media” remove “a”
Line 333 misspelling: endosomes
Response: We have incorporated the changes to the lines #133 and #333.
Reviewer 2 Report
Comments and Suggestions for Authors
This study by Sotorilli et al uses a repurposed drug library to identify compounds that alter St Louis encephalitis virus. They identify a handful of compounds that block SLEV induced cytopatheic effect, determine if the compounds are effective in other cell lines and look for decreases in virus titer. They identify two compounds that alter the host cell and result in lower virus titer. The paper is well written and data is clearly presented with appropriate conclusions drawn.
Author Response
This study by Sotorilli et al uses a repurposed drug library to identify compounds that alter St Louis encephalitis virus. They identify a handful of compounds that block SLEV induced cytopatheic effect, determine if the compounds are effective in other cell lines and look for decreases in virus titer. They identify two compounds that alter the host cell and result in lower virus titer. The paper is well written and data is clearly presented with appropriate conclusions drawn.
» Response: We would like to thank Reviewer 2 for reviewing our manuscript and for the positive comments.
Reviewer 3 Report
Comments and Suggestions for Authors
In the manuscript by Sotorilli et al, the authors describe a drug screen for antiviral activity against SLEV. The project is soundly performed and the results are presented well. Below are my suggestions for improvement.
The title states "MMV Covid Box" which is not appropriate. This is not a known term nor is it defined in the title. Furthermore, it suggests to the reader that the paper may address COVID, which it does not. This term should be replaced and the term needs to be defined in the abstract.
Figure 2 - it is not clear how the 9 compounds are identified in panel A. In panel B, why are 10 compounds shown? In panel C, how do the authors narrow down to 7 compounds?
Figure 3 - Panel A is missing statistical analyses. Panel B should show "DMSO" for the 1st column. All are infected with SLEV, not the 1st column only.
Figure 4 - panels A and B are missing statistical analyses.
Figure 6 - this experiment needs quantification of the images presented. As it stands, lines 294-296 of the text are not supported by the images alone. It is not clear to the reader what differences, if any, should be focused on to analyse the results.
Figure 7 - this figure is uninterpretable. What do the circled letter/number combinations represent? What are the "targets" in line 338? How are P-values calculated in panels B-C?
Author Response
In the manuscript by Sotorilli et al, the authors describe a drug screen for antiviral activity against SLEV. The project is soundly performed and the results are presented well. Below are my suggestions for improvement:
- The title states "MMV Covid Box" which is not appropriate.  This is not a known term, nor is it defined in the title.  Furthermore, it suggests to the reader that the paper may address COVID, which it does not.  This term should be replaced and the term needs to be defined in the abstract.
Response: We switched the term “Covid Box” for “an Open Box Library" in the title and added the term “Medicines for Malaria Venture COVID box” in the abstract to clarify that Covid Box is merely the name of the compound library. Additionally, we included the library name in lines #65, #79, #99, #178, #342 and #467.
- Figure 2 - it is not clear how the 9 compounds are identified in panel A.  In panel B, why are 10 compounds shown?  In panel C, how do the authors narrow down to 7 compounds?
Response: Indeed, compounds Imatinib, Pexidartanib and SAX-187 are missing in the selection of compounds for confirmation and validation. We incorporated a new paragraph on section 3.1 (lines #192 to #198) explaining that SAX-187 required a special license for purchase, which made resupply and further testing unfeasible. Imatinib and Pexidartanib have been identified as false positive hit compounds on our previous screening campaigns using Vero CCL81 cells, and being compounds of the same chemical class and function (tyrosine kinase inhibitors), were excluded from further analysis.
We modified panel 2A to include compounds names and highlight the cut-off parameters for hit compound selection, and changed the color of 7DMA on panel 2B to emphasize that 7DMA was included as a positive control. Panel 2B shows 10 compounds in total, corresponding to 9 hit compounds selected from the screening of the MMV COVID Box library plus 7DMA (positive control). We also changed Figure 2 legend to explain that panel 2C contains only the values of confirmed hit compounds.
- Figure 3 - Panel A is missing statistical analyses.  Panel B should show "DMSO" for the 1st column.  All are infected with SLEV, not the 1st column only.
Response: We replaced the term “SLEV” by “DMSO” as suggested. Regarding the statistical analysis of figure 3A, we analyzed the data again using Kruskal-Wallis test and Dunn's multiple comparisons tests to confirm that the cell viability values in Posaconazole-treated group and the DMSO-treated group are statistically different and added the * symbol for statistical difference.
- Figure 4 - panels A and B are missing statistical analyses.
Response: We included the missing statistical analysis to figure 4.
- Figure 6 - this experiment needs quantification of the images presented.  As it stands, lines 294-296 of the text are not supported by the images alone.  It is not clear to the reader what differences, if any, should be focused on to analyse the results.
Response: We removed figure 6 and associated text from the manuscript, as we realized from the reviewers comments that data was incomplete and added little to the manuscript.
- Figure 7 - this figure is uninterpretable.  What do the circled letter/number combinations represent?  What are the "targets" in line 338?  How are P-values calculated in panels B-C?
Response: We updated Figure 6 [Figure 7 in the older version] to improve clarity:
Circle-Shaped Nodes and Targets:
The Uniprot names on the circle-shaped nodes have been replaced to gene names (acronyms) to improve clarity and understanding. The legend has also been revised to provide a more explicit explanation of the interaction network.
P-Values Calculation in Panels B-C:
In panels B and C, the p-values were obtained from the DAVID platform. The calculation is based on Fisher's Exact test, which assesses gene enrichment in annotation terms when two independent groups can fall into two exclusive categories. Specifically, the p-values are derived to address the question of whether the proportion of genes in the list related to the pathway is significantly different from what would be anticipated based on the background prevalence in the entire genome.
Results and Discussions:
The results and discussions related to the figure have been revised to provide a more comprehensive and clear interpretation of the findings.
Round 2
Reviewer 1 Report
Comments and Suggestions for Authors
My major concerns have been addressed. The manuscript has been edited and the goal of the paper is now better clarified as a proof of concept. I confirm, this study represents a novel screening approach which will be beneficial for the scientific community.
Reviewer 3 Report
Comments and Suggestions for Authors
Thank you for the careful revision.